# Vitamin E Deficiency Disrupts Gene Expression Networks during Zebrafish Development

**DOI:** 10.3390/nu13020468

**Published:** 2021-01-30

**Authors:** Brian Head, Stephen A. Ramsey, Chrissa Kioussi, Robyn L. Tanguay, Maret G. Traber

**Affiliations:** 1Linus Pauling Institute, Oregon State University, Corvallis, OR 97331, USA; brian.head@oregonstate.edu; 2Molecular and Cellular Biology Program, Oregon State University, Corvallis, OR 97331, USA; 3Department of Biomedical Sciences, College of Veterinary Medicine, Oregon State University, Corvallis, OR 97331, USA; stephen.ramsey@oregonstate.edu; 4School of Electrical Engineering and Computer Science, Oregon State University, Corvallis, OR 97331, USA; 5Department of Pharmaceutical Sciences, College of Pharmacy, Oregon State University, Corvallis, OR 97331, USA; chrissa.kioussi@oregonstate.edu; 6Department of Environmental Toxicology, College of Agricultural Sciences, Oregon State University, Corvallis, OR 97331, USA; Robyn.Tanguay@oregonstate.edu; 7School of Biological and Population Health Sciences, College of Public Health, Oregon State University, Corvallis, OR 97331, USA

**Keywords:** VitE, vitamin E, E+, VitE sufficient, E–, VitE deficient, hpf, hours post-fertilization, α-TTP, α-tocopherol transfer protein.

## Abstract

Vitamin E (VitE) is essential for vertebrate embryogenesis, but the mechanisms involved remain unknown. To study embryonic development, we fed zebrafish adults (>55 days) either VitE sufficient (E+) or deficient (E–) diets for >80 days, then the fish were spawned to generate E+ and E– embryos. To evaluate the transcriptional basis of the metabolic and phenotypic outcomes, E+ and E– embryos at 12, 18 and 24 h post-fertilization (hpf) were subjected to gene expression profiling by RNASeq. Hierarchical clustering, over-representation analyses and gene set enrichment analyses were performed with differentially expressed genes. E– embryos experienced overall disruption to gene expression associated with gene transcription, carbohydrate and energy metabolism, intracellular signaling and the formation of embryonic structures. mTOR was apparently a major controller of these changes. Thus, embryonic VitE deficiency results in genetic and transcriptional dysregulation as early as 12 hpf, leading to metabolic dysfunction and ultimately lethal outcomes.

## 1. Introduction

Vitamin E (VitE) is a potent lipophilic antioxidant and is localized in membranes to protect against lipid peroxidative damage [1]. VitE contributes to membrane fluidity via protection of oxidizable lipid and stabilizes membrane domains to assist cell signaling cascades dependent on membrane protein–protein interactions and ion permeability [2]. VitE must be provided in the diet because it is produced only by photosynthetic organisms, where it accumulates in plant seeds and early embryonic structures [3]. Although VitE was discovered as a dietary component necessary to prevent fetal resorption in rats [4], the molecular basis for this requirement still remains under investigation.

The vertebrate embryonic VitE requirement is time-dependent and is needed between embryonic days (E) 9.5 and 11.5 in rats [5], a developmentally similar period to that of zebrafish between 12 and 24 h post-fertilization (hpf) [6,7]. During this window, the zebrafish embryo undergoes significant growth, nearly tripling in length [8]. This period coincides with segmentation of the mesoderm into somites, notochord vacuolation, and primary and secondary neurulation resulting in brain regionalization and neural tube expansion. We showed both by (1) using morpholinos to block the embryonic translation of the mRNA for the α-tocopherol transfer protein (α-TTP) [9] and by (2) evaluating neuronal structures in VitE-deficient embryos (E–) [10,11,12] that VitE is essential during the early stages of zebrafish neurogenesis. Specifically, blocking TTP mRNA translation with an oligonucleotide in zebrafish was 100% lethal by 24 hpf with noticeable impairment of brain and eye development beginning at 12 hpf [9]. VitE deficiency caused by defective lipoprotein metabolism also produces neurodevelopmental defects in mice, including neural tube defects [13] and exencephaly [14]. Additional neurologic impairments have been reported in *Ttpa^-/-^* mice, which show degeneration of cerebellar Purkinje [15] and spinal cord neurons [16].

In addition to morphologic derangements, E– zebrafish embryos also experience lethal dysregulation of glycolytic metabolism in the first 120 hpf [11], which could be remedied by glucose supplementation at 24 hpf. Notably, E– embryos experience a hypermetabolic state at 24 hpf, which switches to a hypometabolic state by 48 hpf. Importantly, carbohydrate metabolism is necessary to provide precursor molecules for the rapid growth of zebrafish embryos especially between 0 and 48 hpf [17]. In addition, E– embryos between 24 and 48 hpf experience errors in amino acid metabolism and generation of precursor ribonucleotides, tricarboxylic acid (TCA) cycle intermediates and demonstrate over-production of free saturated fatty acids [11,18]. E– embryos by 12 hpf experience increased betaine concentrations, suggesting that this methyl donor is needed for the methionine cycle to maintain S-adenosyl methionine production [19]. These data suggest that epigenetic regulation may be impacted by VitE deficiency. Lee et al. showed that blocking folic acid metabolism and thus preventing methyl donor recycling causes also neurologic defects in zebrafish embryos as early as 10 hpf [20].

Zebrafish development is coordinated by temporal and spatial gene expression networks. Transcriptional regulation is a key component of embryonic fate [21]. The mechanistic target of the rapamycin (mTOR) complex senses energy status, amino acid abundance and redox imbalance to regulate cell survival and proliferation [22,23]. Additionally, in zebrafish, mTOR signaling is necessary in myelination pathways throughout development of the central and peripheral nervous system [24]. VitE deficiency may disrupt metabolic pathways regulated by and restored by mTOR signal integration. Metabolic dysfunction is likely the cause and consequence of transcriptional network alterations that regulate metabolic needs for rapid growth and development [25]. Thus, the VitE deficient state may have marked effects on temporal gene expression patterns in the zebrafish embryo specifically during neural tube formation and brain development. The objective of this study was to identify transcriptional targets of VitE deficiency in embryo development longitudinally. Due to early disruption of the metabolic state in E– embryos [11,18], we examined changes prior to 24 hpf. We hypothesize that VitE is necessary to protect zebrafish transcriptional networks associated with metabolism, cell signaling and embryo energy status. In addition, we sought to identify transcription factors necessary for zebrafish embryo nervous system development that are VitE-dependent and produce the morphological defects previously observed [12,19].

## 2. Methods

### 2.1. Zebrafish Husbandry

All experimental protocols and methods were carried out in accordance with the animal use and care protocol (# 5068) approved by the Institutional Animal Care and Use Committee at Oregon State University. Tropical 5D strain zebrafish were reared in the Sinnhuber Aquatic Research Laboratory at Oregon State University under standard laboratory conditions of 28 °C on a 14 h light/10 h dark photoperiod according to standard zebrafish breeding protocols [8]. At 55 days post-fertilization (dpf), adult zebrafish were randomly allocated to two experimental diets, vitamin E deficient (E–) or sufficient (E+), as described [10].

VitE was extracted from the diets prior to feeding and assayed by HPLC-UV prior to feeding, as described [26]. In the E+ diet, α- and γ-tocopherols were 361 ± 10 and 2.6 ± 0.0 mg/kg ± SEM (*n* = 3 measurements), respectively. In the E– diet, α- and γ-tocopherols were 1.1 ± 0.0 and 0.5 ± 0.0 mg/kg (*n* = 3 measurements), respectively. E– and E+ embryos were obtained by group spawning of adult fish fed either E– or E+ diets for a minimum of 80 days up to 9 months. Embryos were collected, staged and incubated until use in standard embryo media (15 mM NaCl, 0.5 mM KCl, 1 mM MgSO_4_, 0.15 mM KH_2_PO_4_, 0.05 mM Na_2_HPO_4_, 1 mM CaCl_2_, NaHCO_3_ in fish system water, (EM)). Embryos at each time point (12, 18, 24 hpf) were staged according to the appropriate developmental landmarks [8]; embryos at 12 hpf with the presence of 6 somites, 18 hpf with the presence of 18 somites and defined otic vesicle, and at 24 hpf (prim-5 stage) upon first pigment formation in the retinal epithelium and presence of 30 somites. To evaluate transcriptional differences based solely on VitE status, only embryos identified as morphologically normal and at the appropriate developmental landmarks were used for analysis. Embryos were euthanized with tricaine (MS222, ethyl 3-aminobenzoate methanesulfonate salt, Sigma Aldrich) in accordance to animal care and use guidelines.

### 2.2. RNA Extraction and Sequencing

RNA was extracted from embryos staged developmentally at 12, 18 and 24 hpf, pooled (*n* = 10 embryos/sample) and homogenized, with 4 sample pools per VitE status. RNA integrity (9.02 ± 0.15, *n* = 24) was assessed using the 2100 BioAnalyzer Instrument (Agilent) at the Center for Genome Research and Biocomputing at Oregon State University (Appendix A). RNA libraries were prepared with the Lexogen QuantSeq 3′mRNA Seq Library Prep Kit-FWD kits (Lexogen, Vienna, Austria). After library preparation, samples were pooled and sequenced using single-end sequencing with 100 bp reads on an Illumina HiSeq3000 instrument (Illumina, San Diego, CA, USA).

### 2.3. Data Deposition

RNASeq data for this study has been deposited in Gene Expression Omnibus with accession number GSE164848 and can be viewed at: https://www.ncbi.nlm.nih.gov/geo/query/acc.cgi?acc=GSE164848.

### 2.4. Data Processing and Statistical Analysis

Data processing was performed using the default parameters provided by the program manuals unless stated otherwise. Sequence read quality was first assessed using FastQC v0.11.5. Adapters and poly(A) tails were trimmed using bbduk v35.92. Trimmed reads were then aligned to the current reference genome (GRCz11, *D. rerio*) with current annotations (Ensembl 100) using STAR aligner, v2.7.3a. Raw gene counts were summarized with edgeR, Bioconductor v3.24.3 in R, v4.0.2. Data were normalized by the TMMwsp (Trimmed Mean of M-values with singleton pairing) method.

Multidimensional scaling plots (MDS) were computed to assess the similarity of group effects, identified as the interaction of diet and time, and the sequencing lane effects between samples. edgeR generalized linear model (glm) was used to fit the data and estimate the common, trended and tagwise dispersions given the experimental design. A quasi-likelihood F-test (QLF) was used to compute differential gene expression between E+ and E– embryo gene expression at each time point measured (*n* = 3 time points; 12, 18, 24 hpf). Heatmaps were generated using *z*-score transformed counts per million (CPM) of differentially expressed (DE) genes identified by adjusted *p*-value, or Benjamini–Hochberg False Discovery Rate (FDR) < 0.1.

### 2.5. Gene Ontology and Network Enrichment

DE genes were subject to Gene Ontology (GO) enrichment analysis using the Bioconductor R package GOseq, v.1.40.0. Over representation analysis (ORA) was performed with DE genes and DE genes with absolute value log_2_FC ≥ 1 found at all time points using WebGestalt [27]. Gene set enrichment (GSEA) was performed with pre-ranked genes computed with log_2_FC and *p*-value using WebGestalt. Differentially expressed transcription factors (TFs) were identified and described by predicted anatomical regions using ZFIN [28]. All network enrichment was considered significant if *FDR* < 0.25. Metabolomic data from 24 hpf embryos previously acquired [11] were integrated with 24 hpf transcriptomic profiles using MetaboAnalyst [29].

### 2.6. Western Blotting

Protein was extracted from pooled (*n* = 30) E+ and E– embryos at 24 hpf in RIPA buffer supplemented with protease and protein phosphatase inhibitors (Calbiochem, La Jolla, CA, USA). Rat liver extract was used as a positive control for the assay. Extracted protein (25 μg) was subjected to SDS-Page and processed by immunoblotting. All gels and blots were run simultaneously. The proteins were visualized by SuperSignal^TM^ West Pico Chemiluminescent Substrate (ThermoFisher, Carlsbad, CA, USA) and quantified using the Bio-Rad Image System (Hercules, CA, USA). The information for antibodies is listed in the Appendix A. Phospho-protein (p-) to unphosphorylated protein ratio is calculated for the representative lanes shown.

## 3. Results

Morphological deformities including yolk sac and pericardial edemas, bent axes, and apparent developmental delays generally occur after 24 hpf in E– embryos as previously reported [12]. To evaluate gene expression differences based on VitE status alone and not phenotypic changes, only embryos identified as morphologically normal were used. All appropriate developmental landmarks were used for the following analyses. Thus, the transcriptional profiles reported herein are powerful determinants of the underlying effects of VitE deficiency because they are not changes induced by physical deformities that induce further stress on the animal.

### 3.1. Hierarchical Clustering and Gene Annotation

To determine the VitE-dependent effects on global transcriptomic changes during embryo development, RNA was isolated from E+ and E– whole embryo lysates (*n* = 4 pools, 10 embryos/pool) at the three time points, 12, 18 and 24 hpf. Multidimensional scaling analysis indicated strong separation between the gene expression profiles by time point and further by embryo VitE status (Appendix A). Sequencing lane effects were similarly analyzed but with little to no effect on gene expression profiles, thus are disregarded from further analysis. Of the 22,796 sequenced genes, 2355 were identified as different (*p* < 0.05) across all time points and by VitE status. From this selected group of genes, 12% (*n* = 285) were considered differentially expressed (DE) under a false discovery rate threshold (*FDR* < 0.1). A heatmap of DE genes was clustered with *z*-score averages according to VitE status, age and gene (Figure 1). Gene clusters from the heatmap were annotated for gene ontology (GO) terms associated with biological processes (BP), molecular function (MF) and cellular component (CC). Cluster 1, identifiable by reduced gene expression in E– embryos at 12 hpf, was significantly (*FDR* < 0.25) associated with organic acid binding (GO:0043177), iron ion binding (GO:0005506), and carbohydrate binding (GO:0030246). Vitamin binding (*FDR* = 0.06, GO:0019842) refers to genes associated with L-ascorbic acid binding capacity, including *egln2*, *p4ha1b*, and *plod1a*. Cluster 2 was significantly associated with peptidase activity (GO:0008233) and unfolded protein binding (GO:0051082). Cluster 3, identifiable by increased gene expression in E+ embryos at 12 and 18 hpf, was significantly associated with protein-containing complex binding (GO:0044877), cell adhesion molecule binding (GO:0050839) and cytoskeletal protein binding (GO:0008092). Cluster 4 genes were defined by GO terms vacuole (GO:0005773) and extracellular region part (GO:0044421). The term “vacuole” was associated with endolysosomal trafficking-associated genes *abcc6a*, *galca*, and *zgc:110239*.

Overall, in E– relative to E+ embryos, there were 110 DE genes with increased expression levels (Figure 2A) and 80 DE genes with decreased expression levels (Figure 2B). DE genes that were observed consistently with increased expression in E– embryos were associated significantly with the following KEGG (Kyoto Encyclopedia of Genes and Genomes) pathways—glycolysis/gluconeogenesis, carbon metabolism, biosynthesis of amino acids, pentose phosphate pathway and fructose and mannose pathways (*FDR* < 0.25, Figure 2C). DE genes that were observed consistently with decreased expression in E– embryos were associated significantly with pyrimidine metabolism and proteasome pathways (Figure 2D).

### 3.2. Top Differentially Expressed Genes in E– Embryos

The top 10 DE genes (e.g., with the largest log-fold changes) with increased or decreased expression levels in E– embryos relative to E+ embryos were identified at each of the three time points (12, 18 or 24 hpf; Table 1). DE genes that increased at all three time points in E– embryos include *olfce2* (olfactory receptor C family, e2), which is predicted to be involved in G protein-coupled receptor signaling, and *cngb3.2* (cyclic nucleotide gated channel subunit beta 3, tandem duplicate 2), which is predicted to have cGMP binding activity and is involved in cation transmembrane transport. DE genes that decreased at all three time points in E– embryos include *serpina7* (serpin peptidase inhibitor, clade A, member 7), a serine-type endopeptidase inhibitor implicated in mTOR-related neuropathology in zebrafish [30] and numerous unnamed genes *si:dkey-73p2.2*, *si:dkey-15j16.6*, and *si:dkey-23k10.3*. *nitr3r.1l* (novel immune-type receptor 3, related 1-like) was found initially (12 and 18 hpf) to be increased highly in E– embryos, but was decreased highly at 24 hpf.

To evaluate the gene product properties of highly DE genes identified, over-representation analysis was performed using DE genes with absolute value log_2_FC ≥ 1, or 2-fold differences between E– and E+ gene expression values. Top GO terms (*FDR* < 0.05) are shown in ascending *p*-value order (Figure 3). DE genes in E– embryos annotated for Biological Process (BP) indicated highly related metabolic processes, including the generation of precursor metabolites and energy and carbohydrate metabolic processes (Figure 3A). The Molecular Function (MF) domain annotations included oxidoreductase activity, glutamate and peptide receptor activity and binding related to cell cycle progression (Figure 3B). Annotations associated with Cellular Component (CC) included further description of energy generation by the respiratory chain complex and other membrane protein complexes (Figure 3C).

Genes (*n* = 22,796) were ranked by their fold-change in expression level between E– and E+ sample groups. Using the ranked genes, Gene Set Enrichment Analysis (GSEA) was carried out using WebGestalt (Figure 4). All GSEA enrichments shown are significant (*FDR* < 0.25). At 12 hpf, E– vs E+ embryos have increased expression of genes associated with vitamin C binding, demethylation and demethylase activity, metabolic processes and decreased expression of genes associated with collagen synthesis and protein import (Figure 4A). At 18 hpf, E– vs E+ embryos have increased expression of genes associated with vitamin C binding and decreased expression of genes associated with oxidoreductase activity, gene expression regulation, endothelium development and membrane protein transport activity (Figure 4B). At 24 hpf, E– vs E+ embryos have increased expression genes associated with neuron ensheathment activity and decreased expression of genes associated with gene transcription regulation (Figure 4C).

Metabolomics data, previously acquired [11], were integrated with the gene expression profiles reported herein (both data sets were from 24 hpf E+ and E– embryos) and submitted to MetaboAnalyst. VitE status significantly (*FDR* = 2.39 × 10^−28^) altered the mTOR signaling pathway (Figure 5A), as shown with metabolites (circles) and genes (squares), which were either increased (red) or decreased (green) in E– relative to E+ embryos. This finding was also validated by Western blot analysis of protein extracts obtained from 24 hpf E– and E+ embryos using β-actin to quantify protein levels and the phosphorylated-to-unphosphorylated protein ratio calculated (Figure 5B). Raptor, a key component of mTOR complex 1 (mTORC1) was decreased in E– embryos. The phosphorylated protein (p-) abundance indicates active or inactive state relative to the unmodified protein, depending on the protein. In E– relative to E+ embryos, at 24 hpf p-Rps6 kinase (p-Rps6k) was decreased more than two-fold, p-eukaryotic translation initiation factor 2A (p-Eif2a) was decreased two-fold, and p-eukaryotic translation initiation factor 4E-binding protein 1 (p-Eif4ebp1) was increased 1.5-fold. These outcomes all support the finding that mTOR signaling was dysregulated at 24 hpf in E– embryos in comparison to E+ embryos.

Over-representation analysis was used for transcription factor (TF) enrichment to determine the extent to which gene expression trends are associated with similar TF regulatory units or promoter regions. DE genes shared the consensus site for TF binding YGTCCTTGR motif (*n* = 12 genes, *FDR* = 0.029). Top DE TFs were identified from the dataset (Table 2). The most DE TF, *nr2f1b,* is expressed in the midbrain hindbrain boundary region and somites of the zebrafish embryo during segmentation [31]. Other highly DE TFs include *mafaa*, expressed in the neural tube and myotomes during segmentation [32], and *zgc:101562*, orthologous to human ZCAN10, ZNF398 and ZNF777. Due to VitE’s role in prevention of propagation of lipid peroxidation, other genes of interest including *ttpa*, *ttpal*, *gpx4a*, *gpx4b* and others were evaluated but were not found to be differentially expressed.

## 4. Discussion

The outcomes of this study show that VitE deficiency disrupts numerous gene expression networks, including energy metabolism, oxidoreductase activity, intra- and intercellular signaling, and developmental transcriptional regulation, during critical developmental windows in zebrafish embryos. We previously identified that VitE deficiency prevents proper neural crest cell migration and impairs midbrain–hindbrain development [12].

E– zebrafish embryos at 24 hpf experience metabolic dysfunction with reduced glycolytic intermediates and increased basal oxygen consumption rates [11]. These metabolic changes are echoed by the gene expression profiles of E– embryos between 12 to 24 hpf reported herein. Similarly, rat embryos between E10.5 and E12.5, developmentally comparable to zebrafish embryos at 24 hpf [7,8], exhibit increased expression levels of genes involved in glycolytic pathways, presumably to increase production of reducing equivalents and precursor molecules via the pentose phosphate pathway. [33]. We found that the greatest changes in gene expression in E– embryos were metabolic processes that generate NADPH, ATP and other precursor compounds through glycolysis, the TCA cycle, and oxidative phosphorylation gene expression networks. Tixier et al. showed that glycolytic pathway induction is required to fuel the rapid growth of zebrafish embryos between 0 and 48 hpf [17]. Glycolysis-related gene expression in E– embryos at 12, 18, and 24 hpf is further supported by increased expression of transcription factor *tead1b* (Table 2, *FDR* = 0.024). Glycolytic genes are necessary for anabolic growth and key morphogenesis pathways in embryonic development, which we previously reported are significantly altered in E– embryos [12,19]. E– relative to E+ embryos at 36 hpf had similar misexpression of carbohydrate metabolism related genes in embryos in our previous microarray experiments [34]. We reported that these same TCA cycle and glycolytic gene products were altered using a proteomic profile in adult zebrafish fed VitE-deficient diets [35]. Vitamin E-deficient rat livers were analyzed by NMR and were found also to have decreased glucose levels [36]. Thus, VitE deficiency appears to impair production of energy.

Metabolic pathways are also critical for cellular compensation in response to lipid peroxidation (LPO). We reported previously the depletion of long chain polyunsaturated fatty acids, specifically docosahexaenoic acid (22:6n-3), in the E– embryos [18] and E– adult zebrafish [37]. Additionally, an LPO biomarker, malondialdehyde (MDA) was increased in E– adult zebrafish [38]. LPO requires detoxification by oxidation of glutathione (GSH). To replenish GSH requires synthesis using the limiting amino acid, cysteine. We have found in E– embryos that choline is depleted, betaine is increased [19] and the methyl donor status [19], along with the thiol status, is disrupted. In addition, oxidized glutathione (GSSG), which requires enzymatic reduction using NAD(P)H, is increased [19]. Thus, the E– embryo, over time, is unable to compensate for the increasing LPO.

*Zgc:101562* and *mafaa*, two differentially expressed transcription factors in E– embryos, are associated with gene expression related to glucose regulation and are expressed in mouse neural tissue at E12.5 [28,39,40] and the zebrafish neural tube during the segmentation period (between 10 and 24 hpf) [28,38,39], respectively. Together, key TFs dysregulated by VitE deficiency, associated with metabolism and localized to early neurologic structures of the vertebrate embryo, highlight VitE’s necessary role in maintaining growth pathways and preventing neurodevelopmental defects we reported previously [12]. Importantly, we showed that glucose injections at 24 hpf rescued ~70% of E– embryos [11], providing a clue that earlier intervention may alter transcriptional responses and fully rescue E– animals through remediation of glycolysis. Importantly, vertebrate neurogenesis formation requires mTOR signaling [41,42] and mTOR is expressed in the zebrafish trunk and brain at 24 hpf [43].

Overall, genes associated with metabolism and mTOR signal integration, used to regenerate the metabolic intermediates described, appear to be the mechanism for the compensatory responses to VitE deficiency. mTOR senses energy and amino acid status signals to stimulate cell survival, growth and proliferation [22]. mTOR is reported also to sense redox status through modulation of glutathione status in the mouse cerebellum [23]. The alteration in mTORC1 at 24 hpf reported herein show that VitE deficiency has a major impact on this key regulator. Additional studies are needed to evaluate the longitudinal changes, along with quantitation of the key mediators, which are likely to accompany the depleted glucose status, oxidized lipids and other cellular stressors.

mTOR contributes to ensheathment in the zebrafish embryo [24] and requires cholesterol for myelination, cytoskeletal support and membrane-specific signaling pathways [44]. We report herein that E– embryo transcripts are enriched for cytoskeleton filament, anchored membrane components and experience metabolic disruption sensed by mTOR. Our data indicate that mTOR signaling is indeed disturbed in E– embryos by 24 hpf with a 3-fold reduction of phosphorylated Rps6k and 2-fold reduction of phosphorylated Eif2a relative to E+ embryos. In addition, expression of genes associated with ensheathment of neurons was positively enriched in E– embryos at 24 hpf. Previously, we showed that a VitE deficiency impacts neural crest cell gene expression patterns and somite integrity as early as 12 hpf. Myelination and ensheathment pathways immediately follow the migration of sensory axons away from the central nervous system (CNS) towards the periphery [45], a process dependent on neural crest cells [46,47]. Hypothetically, the errors at 12 hpf may precede the effects observed in 24 hpf E– embryos. Sensory neurons are derived from Sox10-expressing neural crest cells [48], which we previously reported to be reduced in number and mis-localized in E– embryos at 12 and 24 hpf [12]. Thus, we suggest that VitE deficiency may disrupt mTOR signaling cascades and subsequent ensheathment pathways that begin during axonal migration from the CNS at 24 hpf.

Many other pathways were impacted by VitE deficiency, of note is collagen remodeling. *egln3*, expressed in the neural plate in 12 hpf zebrafish embryos [32], is a hypoxia inducible factor (HIF) target and regulates the expression of collagen remodeling genes *p4ha1b*, *plod1a* and *col1a1b* [49,50]. VitE radicals produced as a result of lipid peroxidation reactions are chemically reduced by ascorbic acid (Vitamin C (VitC)), thereby depleting this critical antioxidant micronutrient [51]. Zebrafish, like humans, likely do not synthesize VitC and require it in their diet [52,53]. VitC was increasingly depleted in E– relative to E+ embryos up to 120 hpf [11]. We observed herein that vitamin binding, annotated primarily with VitC-associated gene expression, was positively enriched in E– embryos at 12 and 18 hpf, indicating VitC’s important early role in development [54,55,56]. VitC is required for collagen prolyl hydroxylation, catalyzed by the *p4ha* family of genes, including *p4ha1b* found in this study [57]. Together, a VitE deficiency and induced VitC deficiency can dramatically alter collagen remodeling pathways in zebrafish embryo vascularization, skin growth and even axonal projections of the brain [58]. Negative enrichment of genes associated with collagen trimer at 12 hpf and endothelium development at 18 hpf in E– embryos further highlights VitE’s role in maintaining structural support during embryonic development of the notochord and somites [59].

In addition, there were numerous genes differentially expressed at 12, 18 and 24 hpf in E– embryos that might suggest multiple developmental pathways disturbed, including—*nr2f1b*, a transcription factor decreased in E– embryos at 12 hpf (log_2_FC = −2.21, *FDR* = 0.007) that regulates vascularization and hindbrain regionalization in zebrafish [60]; *nitr3r.1l*, an immune receptor significantly increased in E– embryos at 12 and 18 hpf (log_2_FC = 6.14 and 2.06, respectively) but decreased at 24 hpf (log_2_FC = −2.31) [61]; and *cngb3.2*, a cation transmembrane transporter increased in E– embryos at 18 and 24 hpf (log_2_FC = 2.94 and 1.59, respectively) implicated in the pathology of retinitis pigmentosa [62], a symptom of VitE deficiency in humans [63]. We also compared our data to that of other VitE deficiency models. Unfortunately, comparisons between species including the mouse, rat, and horse, between tissue types, and by age of development are limited. Thus, we found little to no overlap between our dataset and that of others published [64,65,66,67]. It is most likely that the zebrafish embryo prior to 24 hpf simply does not share similar gene expression profiles of highly differentiated tissues such as the cerebellum, spinal cord or liver. Myelination pathways, for example, are not highly expressed in early development and thus are not differentially expressed in the E– embryo.

In summary, we show that VitE deficiency as early as 12 hpf disrupts pathways underlying growth and development. This study also points to major metabolic dysfunctions occurring as early as 12 hpf. Simultaneously, E– embryos experience LPO disruption to membrane structure and signaling pathways, both of which are known secondary effects due to VitE deficiency. Errors occurring as early as 12 hpf signal catastrophic decline and impairment to neurodevelopment that relies on early signaling capacity for cellular migration, proliferation and overall tissue and organ development. There remains a need to further investigate the origins of VitE deficiency-induced embryonic death as a means of preventing developmental defects and improving prenatal health span.

## Figures and Tables

**Figure 1 nutrients-13-00468-f001:**
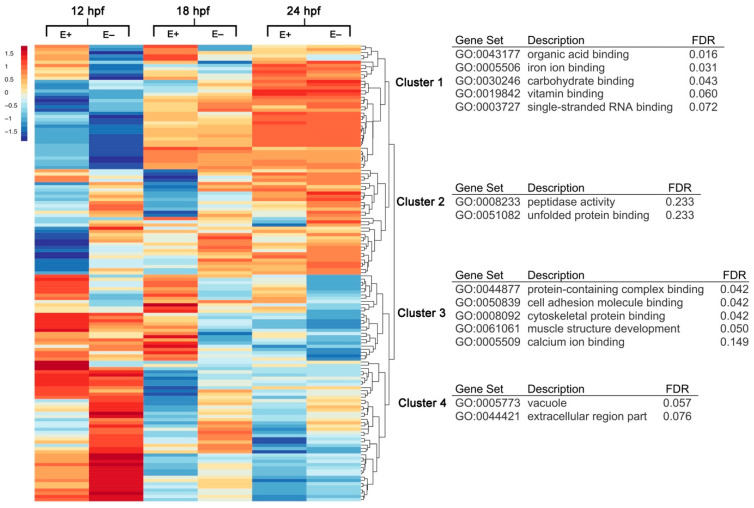
Hierarchical clustering and gene annotation of differentially expressed genes in E– and E+ embryos at 12, 18 and 24 hpf. Heatmap of all differentially expressed (DE) genes (*n* = 286, *FDR* < 0.1) clustered with *z*-score averages according to VitE status, age and gene with red color indicating greater expression relative to all-condition average and blue color indicating lower expression relative to all-condition average amongst all expression values (rows correspond to genes, columns correspond to conditions). Clusters within heatmap annotated by Gene Ontology (GO) terms organized by Benjamini–Hochberg False Discovery Rate (FDR).

**Figure 2 nutrients-13-00468-f002:**
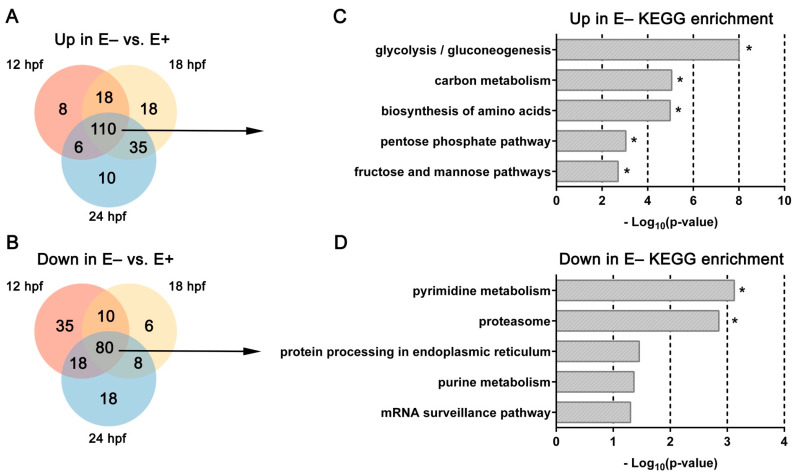
Venn diagrams and KEGG pathways enriched by over representation analysis of differentially expressed genes changed consistently over time. (**A**,**B**) Venn diagrams separating DE genes consistently (**B**) increased or (**C**) decreased in E– embryos relative to E+ embryos at each age. (**C**,**D**) KEGG pathways associated with DE genes (*FDR* < 0.1) consistently (**C**) increased or decreased (**D**) in E– embryos vs. E+ embryos at each time point (12, 18, and 24 hpf). * indicates significant *FDR* < 0.25.

**Figure 3 nutrients-13-00468-f003:**
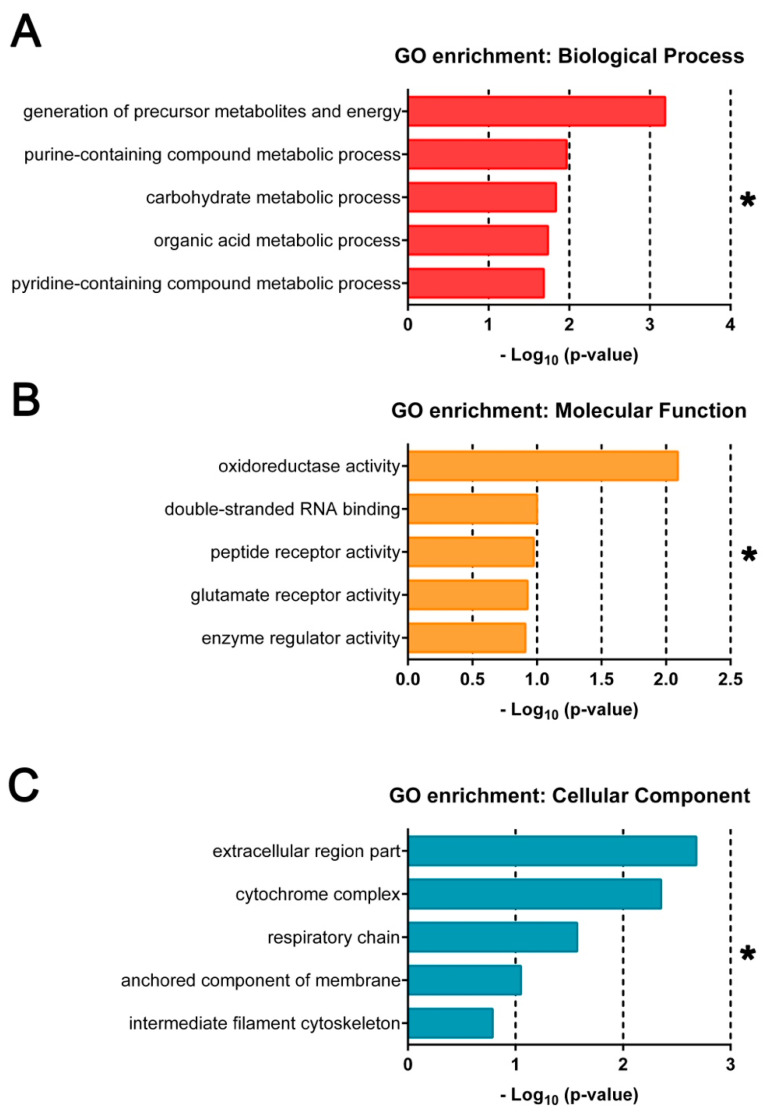
Gene ontology terms enriched by over representation analysis and gene set enrichment analysis of all differentially expressed genes. GO annotations associated with most highly DE genes (*n* = 286, *FDR* < 0.1, |log_2_(FC)| ≥ 1) enriched across all ages (12, 18 and 24 hpf) and Vitamin E (VitE) status by over-representation analysis. Terms are separated by (**A**) Biological Process, (**B**) Molecular Function and (**C**) Cellular Component. * indicates significant FDR < 0.25 with all enrichments shown considered significant.

**Figure 4 nutrients-13-00468-f004:**
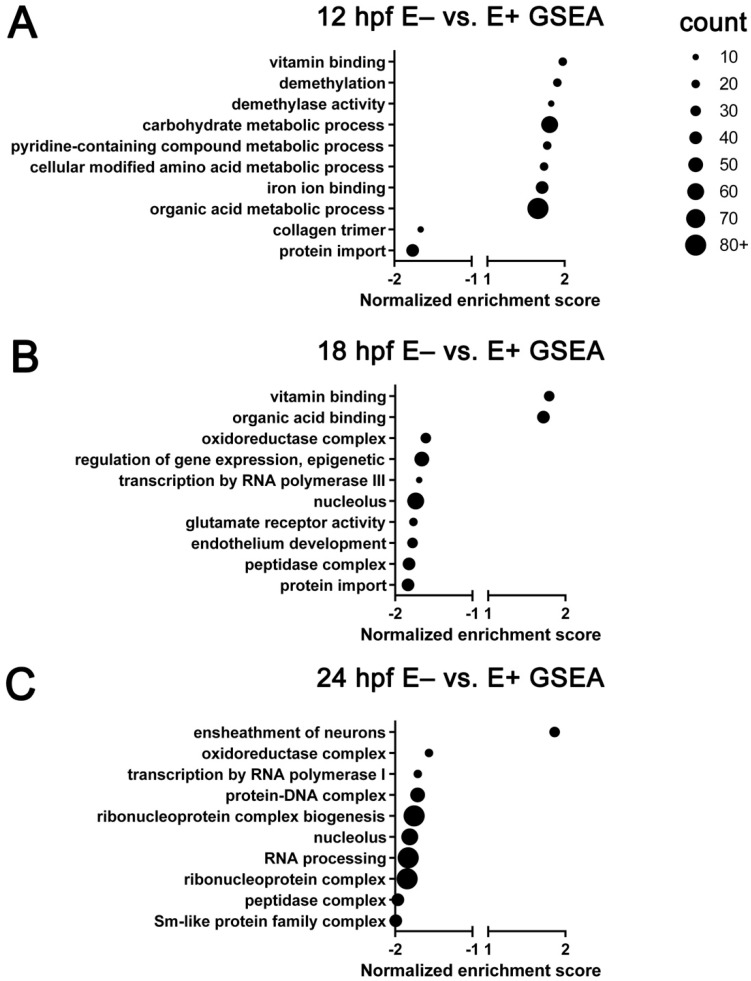
Gene ontology terms enriched by gene set enrichment analysis of all expressed genes. GO annotations associated with all genes (*n* = 22,796) pre-ranked, analyzed by time (12, 18, 24 hpf) and organized by normalized enrichment score. Top 10 significant (*FDR* < 0.25) GO terms are separated by expression at (**A**) 12 hpf, (**B**) 18 hpf and (**C**) 24 hpf. Approximate number of genes found enriched in each term indicated by dot size. Positive enrichment score indicates increased expression in E– relative to E+ embryos, negative enrichment score indicates decreased expression in E– relative to E+ embryos. Normalized enrichment score automatically computed with enrichment and adjusted *p*-value by WebGestalt.

**Figure 5 nutrients-13-00468-f005:**
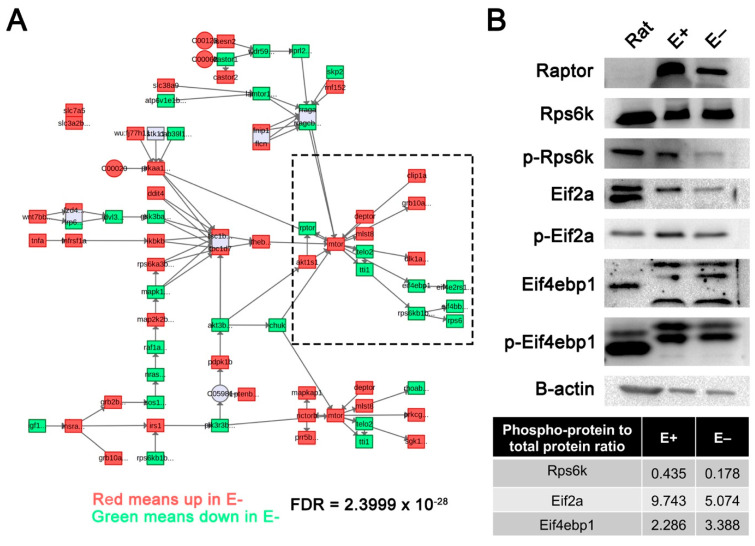
Mechanistic target of rapamycin (mTOR) signaling pathway disrupted in 24 hpf E– embryos. (**A**) E+ and E– embryos at 24 hpf metabolomic data acquired previously were integrated with gene expression profiles using MetaboAnalyst to generate network profiles. The mTOR signaling pathway was significantly enriched with both metabolites and genes expressed in 24 hpf E– embryos. Red boxes or circles represent increased, while green boxes represent reduced metabolite or gene expression, respectively, in E– relative to E+ embryos. (**B**) mTOR complex 1 (mTORC1)-associated proteins including Raptor, Rps6k, Eif2a and Eif4ebp1 were evaluated by activity determined by phosphorylation status in pooled protein extracts of rat liver (positive control), E+ and E– embryos (*n* = 30 embryos/pool) at 24 hpf. Phospho-protein (p-) to unphosphorylated protein ratio is calculated for the representative lanes shown.

**Table 1 nutrients-13-00468-t001:** Top differentially expressed genes in E– embryos relative to E+ at 12, 18 and 24 hpf. Top 10 DE genes by absolute value log_2_FC at each time point with log_2_FC in E– embryos relative to E+. ↑ and ↓ indicate a DE gene found in the top 10 consistently increased or decreased in E– embryos at each age (12, 18, 24 hpf).

12 hpf		18 hpf		24 hpf	
Gene Symbol	Log_2_FC in E–	Gene Symbol	Log_2_FC in E–	Gene Symbol	Log_2_FC in E–
*nitr3r.1l* ↑↑↓	6.129	*cngb3.2* ↑↑	2.944	*olfce2* ↑↑↑	5.822
*si:dkey-90m5.4*	3.135	*olfce2* ↑↑↑	2.472	*si:dkey-23k10.3* ↓↑	4.389
*tnnc1a*	3.048	*ighd* ↑↑	2.326	*si:dkey-238d18.15* ↑↑	3.404
*olfce2* ↑↑↑	3.023	*anpepa* ↑↑	2.129	*polr3c*	2.768
*cd40lg*	2.924	*nitr3r.1l* ↑↑↓	2.068	*topaz1*	2.242
*slc25a55b*	2.496	*si:dkey-238d18.15* ↑↑	2.003	*anpepa* ↑↑	1.815
*ribc1*	2.386	*rnps1*	1.801	*si:dkey-238d18.5*	1.813
*ighd* ↑↑	2.327	*pvalb4*	1.714	*cngb3.2* ↑↑	1.722
*si:dkey-159f12.2*	2.225	*snx21*	1.662	*dntt*	1.516
*scdb*	2.145	*si:dkey-90m5.4*	1.635	*f2rl1.2*	1.505

*b3gat2*	−4.178	*serpina7* ↓↓↓	−4.396	*nitr3r.1l* ↑↑↓	−2.692
*serpina7* ↓↓↓	−2.993	*si:dkey-15j16.6* ↓↓	−2.998	*si:ch211-191i18.2*	−2.657
*loxhd1b* ↓↓	−2.787	*si:dkey-73p2.2* ↓↓↓	−2.901	*si:dkey-15j16.6* ↓↓	−2.648
*si:dkey-73p2.2* ↓↓↓	−2.673	*zgc:158445*	−2.839	*serpina7* ↓↓↓	−2.629
*asb13a.1*	−2.666	*si:ch1073-13h15.3*	−2.680	*si:dkey-73p2.2* ↓↓↓	−2.231
*rfesd*	−2.532	*si:dkey-23k10.3* ↓↑	−2.460	*zgc:173585*	−1.617
*aqp8b*	−2.518	*prkra*	−2.174	*loxhd1b* ↓↓	−1.575
*nr2f1b*	−2.205	*grin1b*	−1.573	*zgc:101562*	−1.453
*proca*	−2.069	*kiaa1549lb*	−1.570	*insl5a*	−1.206
*si:ch1073-268j14.1*	−2.041	*ormdl3*	−1.417	*dock11*	−1.125

**Table 2 nutrients-13-00468-t002:** Top transcription factors differentially expressed in E– embryos across the developmental window measured. Transcription factors (TFs) identified within DEG list annotated for TF family, anatomical region of expression at the measured time point and log_2_FC in E– embryos relative to E+.

Symbol	Name	Log_2_FC in E– vs. E+			FDR
		12 hpf	18 hpf	24 hpf	
*nr2f1b*	Nuclear receptor subfamily 2, group F, member 1b	−2.21	0.17	0.12	0.007
*mafaa*	v-maf avian musculoaponeurotic fibrosarcoma oncogene homolog Aa	−1.83	−0.22	0.13	0.008
*hsf1*	Heat shock transcription factor 1	0.68	0.55	−0.08	0.013
*zgc:101562*	zgc:101562	−0.94	−1.11	−1.45	0.020
*tead1b*	TEA domain family member 1b	0.46	0.26	0.15	0.024
*mbd3b*	Methyl-CpG binding domain protein 3b	−0.32	−0.35	−0.23	0.024
*tox2*	TOX high mobility group box family member 2	−0.48	−0.41	−0.08	0.027
*hmgb2b*	High mobility group box 2b	−0.14	0.49	0.04	0.047
*bhlhe40*	Basic helix-loop-helix family, member e40	0.86	0.17	0.12	0.061
*tbx18*	T-box transcription factor	−0.81	−0.33	−0.42	0.066

## Data Availability

RNASeq data for this study has been deposited in Gene Expression Omnibus with accession number GSE164848 and can be viewed at: https://www.ncbi.nlm.nih.gov/geo/query/acc.cgi?acc=GSE164848.

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
