# Peer review of "Vitamin E Deficiency Disrupts Gene Expression Networks during Zebrafish Development"

_nutrients, 2021, doi:10.3390/nu13020468_

Round 1

Reviewer 1 Report

Review

Tittle: Vitamin E deficiency disrupts gene expression networks during zebrafish development

Authors: Brian Head et al.,

The manuscript at hand examines potential mechanisms of Vitamin E functions on embryonic development in Zebra fish. The investigators use RNASeq and further standard bioinformatic analysis to assess the effects of Vitamin E deficiency on gene expression at different stages of embryionic development. The study reports on specific genetical expression difference observed and discuss potential biological effects.

In general the study reports important findings and the investigators address most significant discoveries in the discussion. My main suggestion to the investigators is to revise the discussion in order to be less categorical in concluding a direct cause-effect relationship in the E- groups. It is important to avoid suggesting very conclusive using the RNASeq Methods due to the possibility of having unrelated  effects. These effects that may be observed in the expression of other genes not linked to the particular pathway affected. The study would have benefitted in also performing analysis of those cluster genes. I have few specific comments.     

  1. Please show data showing analysis of RNA integrity of Vitamin E and Vitamin E deficient cohorts used for the RNASeq analysis.
  2. The findings reported suggest the absence of a cellular compensatory mechanisms to  address vitamin E deficiency. Please comment.
  3. Please expand on the suggestion that Vitamin E deficiency alters ensheathment of neurons during development specifically as it relates with a potential effects on the  MTOR pathway.  
  4. Correct space in text line 202 

Reviewer 2 Report

Vitamin E (vitE) is a micronutrient required for vertebrate embryonic development, as shown by the authors and several other research groups. In the present study, the authors have used RNA-Seq to determine transcriptomic changes associated with vitamin deficiency in zebrafish development and find a number of genes involved in different processes that seem to be regulated by the vitE status of the embryos. The possible significance of these changes is discussed in detail, but no functional data has been provided. The work presented in this study provides an important knowledgebase, with which more mechanistic and functional research can be designed.

The question explored in this manuscript has been studied before by the authors using microarray, but RNA-Seq is a much better approach and therefore this study represents an advance in the field. However, in its current form, the study is very descriptive and lacks functional data to support the importance of the findings. Since vitE status appears to be relevant for mammal neural tube closure (possibly humans), gaining new functional insights is of considerable significance and the zebrafish model facilitates these types of studies. Overall, this study could direct these future studies, but could gain more merit by adding complementary data supporting the authors' claims.

In general, the manuscript is well written and can be easily understood. The methodology is described in good detail, but the publication of the code used for the analyses would be very helpful. This could be done as supplemental material or in a suitable repository (e.g. GitLab). As described in the manuscript, the tools used are generally appropriate for the analysis of RNA-Seq data.

General comments

  1. This study is highly descriptive in nature and does not provide complementary data validating sequencing results or their importance. Authors could perform immunofluorescence for select targets to show their localization in structures of interest (somites, notochord, etc.). In addition, authors could perform some intervention to obtain functional data on select targets: does direct inhibition of glycolisis or oxidative phosphorylation produce a similar phenotype? knock-down of one of the identified transcription factors?

  2. Authors should compare their dataset to other published datasets of vitE deficiency and discuss similarities and differences.

  3. I understand that the authors have extensive experience with this zebrafish model of vitE deficiency and have characterized it in previous work, but it would be helpful to have some morphological data on the specific samples used here. In the methods section, the authors describe the inclusion criteria for the larvae; these data should be added as a figure to have an idea of the morphological derangement that could be cause or consequence of the transcriptional changes.

  4. The use of EdgeR for DE analysis is not an issue, but DESeq2's approach has more power when sample size is small and can also accommodate complex designs into the glm. Can the authors reanalyze the data using DESeq2 to see if they find more DE genes? This could be useful in detecting other attractive targets.

  5. Raw sequencing data should be deposited in a standard repository and a Data Availability statement added to the manuscript.

  6. All DE genes found in the study should be reported in a supplementary table and hopefully in a machine-readable format like .xls.

  7. Is the GSEA implementation in WebGestalt equivalent to the original one from the Broad Institute? The implementation from the fgsea package in R can give different results from the original software, so the authors should check this by analyzing the data with the software from the Broad Institute.

It also would be helpful to add the more classic GSEA plots for the terms described in Fig. 4 as supplementary material to show the rank of the genes driving the enrichment scores.

  1. Supplementary Fig. 3 should be referenced in the results section and not only in the discussion. This data could also be exploited further; what metabolic implications can be extracted from the disruptions in the network? how could they be tested experimentally?

  2. Including mTOR data in the manuscript would help strengthening the arguments. In the current version of the manuscript it is referenced as 'unpublished data', but should be added as a new figure.

Specific comments

  1. Line 128: RIN states an N = 16, but the authors sequenced 24 samples.

  2. Line 139: Authors used STAR for aligning, but the libraries were from the 3' end with UMI. Did the authors use UMI to account for PCR duplicates? if so, how?

  3. As said above, authors could use DESeq2 Wald's test using independent filtering to obtain more power with this small sample size.

  4. Lines 174 - 179: It is not clear from the text what is the purpose of making the heatmaps. This could be clarified better.

  5. The discussion can be made more concise by reducing speculations and focussing on the more important aspects of the study.

Round 2

Reviewer 2 Report

The author's have adequately addressed most of my concerns in the new version of the manuscript; however, some issues remain.

- I understand that posting all the code used in the analysis may be hard, but at least the authors should provide the parameters chosen for each software. If default parameters were used, this should be stated.

- The model used in this and other studies by the authors is heterogenous, as is common in developmental biology in different species, including zebrafish. Morphological descriptions of the samples used in this study specifically would be helpful to assess the data. In the methods, it is stated that embryos were selected based on developmental milestones, but no information is given about other important characteristics. For example, edema or yolk sac defects could be important in driving transcriptional differences in metabolic processes.

- Western blots seem to be upside down for some of the proteins, based in the ladder and the band migration pattern. These features are present in the original scans and might be just a vertical mirro flip, but authors should check this.
